# Effect of Lipids in Yak Muscle under Different Feeding Systems on Meat Quality Based on Untargeted Lipidomics

**DOI:** 10.3390/ani12202814

**Published:** 2022-10-18

**Authors:** Lin Xiong, Jie Pei, Xingdong Wang, Shaoke Guo, Xian Guo, Ping Yan

**Affiliations:** 1Animal Science Department, Lanzhou Institute of Husbandry and Pharmaceutical Sciences, Chinese Academy of Agricultural Sciences, Lanzhou 730050, China; 2Key Laboratory of Animal Genetics and Breeding on Tibetan Plateau, Ministry of Agriculture and Rural Affairs, Lanzhou 730050, China; 3Key Laboratory for Yak Genetics, Breeding and Reproduction Engineering of Gansu Province, Lanzhou 730050, China

**Keywords:** yak, meat quality, flavor, lipid, feeding system

## Abstract

**Simple Summary:**

With the development of living standards, consumers are paying more and more attention to meat quality and flavor. When consumers choose meat, they directly pay attention to meat quality and flavor, so the meat quality and flavor directly decide meat price and sales volume. Better meat quality and flavor are the crucial factors that increase the additional value of meat. Because of its special nutritional value and taste, yak meat is popular with consumers. The intramuscular lipids can greatly affect the meat quality and flavor, but there is no report on the effect of lipids in yak muscle on the meat quality and flavor. In this study, we studied the characterization of lipids in yak muscle under different feeding systems and further explored the key lipids affecting yak meat quality and flavor. This study can provide new insight into the improvement of yak meat quality and flavor.

**Abstract:**

The effect of lipids on yak meat quality and volatile flavor compounds in yak meat under graze feeding (GF) and stall feeding (SF) was explored using untargeted lipidomics based on liquid chromatography–mass spectrometry (LC-MS) in this study. First, the volatile flavor compounds in longissimus dorsi (LD) of SF and GF yaks were detected by gas chromatography–mass spectrometry (GC-MS). In total 49 and 39 volatile flavor substances were detected in the LD of GF and SF yaks, respectively. The contents of pelargonic aldehyde, 3-hydroxy-2-butanone and 1-octen-3-ol in the LD of both GF and SF yaks were the highest among all detected volatile flavor compounds, and the leading volatile flavor substances in yak LD were aldehydes, alcohols and ketones. In total, 596 lipids were simultaneously identified in the LD of SF and GF yaks, and the leading lipids in the LD of both GF and SF yaks were sphingolipids (SPs), glycerolipids (GLs) and glycerophospholipids (GPs). Seventy-five significantly different lipids (SDLs) between GF and SF yaks were identified in the LD. The high content of TG(16:1/18:1/18:1), TG(16:0/17:1/18:1) and TG(16:0/16:1/18:1), PE(18:0/22:4) and PC(18:2/18:0) can improve the a* (redness) and tenderness of yak muscle. The changes in volatile flavor compounds in yak muscle were mainly caused by TG(18:1/18:1/18:2), TG(18:0/18:1/18:1), TG(16:0/17:1/18:1), TG(16:0/16:1/18:1), PC(18:2/18:0), TG(16:1/18:1/18:1), PI(18:0/20:4), TG(16:1/16:1-/18:1) and TG(17:0/18:1/18:1). The above results provide a theoretical basis for improving yak meat quality from the perspective of intramuscular lipids.

## 1. Introduction

In recent years, the consumers’ demands on meat quality are becoming higher and higher, and the improvement of meat quality is one of the most notable subjects in animal husbandry. Lipids are one of the main constituents in livestock muscle and play a key role in physiological actions such as energy storage, hormonal regulation, signal transmission and biofilm formation [1]. The different lipids possess different physiological activities for the human body [2]. For example, sphingolipids (SPs) can prevent some cancer, and glycerophospholipids (GPs) can reduce the risk of cardiovascular and cerebrovascular diseases. On the other hand, lipids are also a main factor affecting meat qualities such as water-holding capacity, juiciness and tenderness [3,4,5,6]. Many studies have been carried out in order to explore the relationship between intramuscular lipids and meat quality. Consumers first pay attention to the meat’s sensory quality, and the meat flavor is the other leading factor in a consumer’s meat choice [7]. Lipids are also important precursors of flavor compounds, and the volatile flavor compounds in meat are mainly affected by lipids [8,9]. The volatile flavor compounds in livestock muscle are mainly produced by lipid oxidation and degradation, and the type and amount of lipids in livestock muscle can cause different flavors [10,11,12]. Breed [13], diet [14], gender [15], age [16] and feeding system [17] can influence the characterization of lipids in livestock muscle. Among the above factors, the feeding system is very essential. In general, the feeding systems for livestock mainly include graze feeding (GF) and stall feeding (SF). GF is likely to change the lipid composition and especially increase the content of unsaturated fatty acids (UFAs) in livestock muscle, whereas SF can improve overall lipid content [18].

Yak (*Bos grunniens*) originated on the Qinghai–Tibet Plateau and possesses strong adaptability to cold and anoxic natural environments in plateau pastoral areas. As a main livestock species on the Qinghai–Tibet Plateau, yak is the dominant means of production and livelihood for plateau herdsmen [19]. Yak meat is one of the main sources of animal protein for local residents, and the annual yield has been approximately 300 thousand tons in recent years. As a kind of green food, fresh yak meat and meat production are becoming more and more popular with consumers [20]. Yak meat possesses plenty of functional lipids, which can be developed into commercial products. However, there is a special flavor in yak meat that can cause consumers to have a negative perception of its smell. As yak is a kind of grazing livestock species, its growth and development are very slow due to the harsh natural environment on Qinghai–Tibet Plateau [19]. In recent years, the SF for yak has been increasingly implemented to change the backward situation of yak production. The change in the feeding system in yak production may lead to differences in lipids in yak muscle, further affecting the volatile flavor compounds in yak meat. As far as we know, there are no reports on the effect of lipids on yak meat quality and flavor. In the last few years, robust analytical techniques have been developed in order to achieve the reliable characterization of lipid profiles in biological samples [21]. Lipidomics is a systems biology approach that can study the lipid molecules in an organism on a system-wide level [22]. This method has become a rapidly expanding research field and has been widely used in exploring lipid characterization in different biological samples [23,24]. In animal husbandry, it has been successfully applied to studying the lipid characterization of dairy cattle [25,26], pork [27], cattle [28,29,30], goat [31] and lamb [32], and plenty of lipid molecules with active functions have been identified. The approach of lipidomics in yak muscle can reveal the comprehensive lipid profile, which not only will explore the characterization of yak intramuscular lipids, but also is important in understanding and screening these lipid molecules associated with yak meat quality and flavor.

In this study, we aimed to explore the effect of lipids on yak meat quality and flavor. The yaks under GF and SF were chosen as the different models of lipids and yak meat quality and flavor. First, the data of the meat quality of the yak longissimus dorsi (LD) under SF and GF were collected, and volatile flavor compounds and lipid profiles were detected by gas chromatography–mass spectrometry (GC-MS) and liquid chromatograph–mass spectrometry (LC-MS) untargeted lipidomics, respectively. The feature of meat quality, lipids and volatile flavor compounds in yak LD under GF and SF were revealed. Further, the correlation of lipid molecules with meat quality and volatile flavor compounds was analyzed, and the crucial lipid molecules affecting yak meat quality and flavor were screened. This study will greatly and positively influence the development of yak industrialization and the improvement of yak meat quality.

## 2. Materials and Methods

### 2.1. Animals and Sample Collection

Twelve healthy male yaks (210.33 ± 10.23 kg, two-year-old) were chosen and randomly divided into a GF group (six yaks were only grazed in a natural pasture with no supplements) and an SF group (six yaks were fed with total mixed ration (TMR) in stalls). The composition of TMR was as follows: corn (19.20%), wheat bran (9.20%), whole corn silage (32.00%), oat hay (28.00%), rapeseed meal (8.10%), NaHCO3 (1.00%), NaCl (1.50%) and premix (1.00%). The nutritional contents in grass and TMR are shown in Appendix A. All yaks possessed the same genetic background; were fed in Qinghai province, China; and were dewormed before the test. The yaks in the same group were given free-choice access to diet and water. After being fed for six months, the final body weights of yaks in the GF and SF groups were 305.37 ± 12.08 and 378.87 ± 10.21 kg, respectively. Twelve experimental yaks were fasted for 24 h according to the animal welfare procedures and then were exsanguinated after being stunned by electricity at a commercial abattoir (Xiahua Meat Food Co. Ltd., Xining, China). The slaughter procedure was conducted in accordance with European Commission Regulation. An LD sample (12th–13th rib level) of about 2 kg was collected from each yak immediately. Part of the LD sample was used to measure meat quality on the scene, and part of the sample was kept in liquid nitrogen for lipidomic analysis. The remaining LD sample was frozen at −80 °C for detecting volatile flavor substances. The animal experiment was approved by the Ethics Committee of the Lanzhou Institute of Husbandry and Pharmaceutical Sciences, Chinese Academy of Agricultural Sciences (Permit No. SYXK-2020-0166).

### 2.2. Determination of Meat Quality and Total Lipid Content

The meat quality was analyzed following the methods described by the Association of Official Analytical Chemists (AOAC). After slaughter, the meat color, pH, shear force, cooking loss and water loss of yak LD were directly measured at the slaughtering factory. Meat color parameters including lightness (L*), redness (a*) and yellowness (b*) were measured at 45 min and 24 h postmortem at 4 °C with a CR-400 chroma meter (Konica Minolta Inc., Tokyo, Japan). After the carcass was cooled, the probe of the chroma meter was placed vertically on the cut surface of yak LD. During the measure, the cut surface of yak LD should not be exposed to the air for more than 10 min, and three measurements were taken from each LD sample. The pH was measured at 45 min and 24 h postmortem with a TESTO 205 pH meter (TESTO AG Inc., Lenzkirch, Germany). The probe of the pH meter was inserted into the LD sample and read after full contact. Shear force was measured with a C-LM4 tenderness meter (Northeast Agricultural University, Shenyang, China). Cooking loss was measured with a water bath at 80 °C. The fascia and fat were removed from the LD sample, and the sample was weighed (W_1_); then, the sample was put in a steamer and boiled for about 30 min. The cooked meat was taken out and cooled for 15 min under natural conditions and then weighed (W_2_). Cooking loss was calculated as cooking loss (%) = W_1_ − W_2_/W_1_ × 100. Water loss was measured by weight loss over 24 h at 4 °C in a plastic bag. Muscle slices about 1.0 cm thick perpendicular to the direction of muscle fibers were used. The lipid content was determined by the Soxhlet extraction method. The LD sample was directly extracted with petroleum ether, and then extracting solution was evaporated. The lipid content in yak LD was obtained and weighed.

### 2.3. Determination of Volatile Flavor Compounds

The volatile flavor compounds in yak LD were detected according to the method described in [33] with some modifications. Headspace solid-phase microextraction (HS-SPME) combined with QP 2010 GC-MS (Simadzu, Kyoto, Japan) coupled with a Thermo Scientific TRACE TR-FFAP capillary column (30 m × 0.25 mm, 0.25 μm; Waltham, MA, USA) was used for analyzing the volatile flavor compounds in yak LD. The yak LD sample was mashed with a BÜCHI Mixer B-400 homogenizer (Flawil, Switzerland), and then 5 g of the mashed sample was placed in a 20 mL headspace bottle. The bottle was put in a fridge at 4 °C. Before use, the extraction head (Supelco, Bellefonte, PA, USA) was held at 220 °C for 30 min in the GC inlet; then, it was inserted into the headspace bottle containing the mashed yak LD sample, and volatile flavor compounds were extracted at 80 °C for 10 min. At last, the volatile flavor compounds were desorbed from the extraction head in the GC inlet for 5 min. The GC conditions were as follows: the carrier gas was high-purity helium (He) and the flow rate was 1 mL/min; the programmed temperature in the column oven was the initial temperature 40 °C, 5 min, 4 °C/min to 120 °C, 10 min, then 13 °C/min to 220 °C, holding for 5 min; the inlet temperature was 220 °C; and the split injection ratio was 20:1. MS conditions were as follows: the transmission line temperature was 220 °C; ion source temperature was 200 °C; ionization mode was electron impact ionization, and ionization voltage was 70 eV; scanning mode was full scan, and scan range was *m/**z* 35–500. The MS data obtained with GC-MS were compared with the NIST online databases, and only compounds with both positive and negative similarity index > 750 were chosen and analyzed. Volatile flavor compounds were confirmed by the qualitative analysis. The relative content of each volatile flavor compound was calculated by the area normalization method. These volatile flavor compounds were classified as aldehydes, ketones, alcohols, esters, aliphatic hydrocarbons, heterocyclic compounds, aromatics and acids. The total content of aldehydes, ketones, alcohols, esters, aliphatic hydrocarbons, heterocyclic compounds, aromatics and acids was the sum of all volatile flavor compounds’ content in each category, respectively.

### 2.4. Determination of Lipidomics

A 50 mg yak LD sample was put in an Eppendorf (EP) tube, and 20 μL Lyso PC17:0 solution in methanol (0.01 mg/mL), 10 μL 2-chloro-l-phenylalanine solution in methanol (0.3 mg/L) and 500 μL pure water were added into the tube in sequence. Two steel balls were placed in the tube, and the mixture was precooled for 2 min at −20 °C and was then ground with a Tissuelyser-48 grinder (Shanghai Jingxin Industrial Development Co., Shanghai, China). Next, 300 μL chloroform was added, and the sample was vortexed for 30 s and extracted with an F-060SD ultrasonic instrument (Shenzhen Fuyang Technology Group Co. Ltd., Shenzhen, China) for 10 min in an ice–water bath and then stored at −20 °C for 20 min. The mixture was centrifuged at 4 °C (13,000 r/min) for 10 min, and then 200 μL lower solution was transferred into a new centrifuge tube. Three hundred microliters of a solution of chloroform–methanol (*v:v*, 2:1) was added into the previous centrifuge tube, and the mixture was vortexed for 30 s, followed by extraction with the ultrasonic instrument once again. Two hundred microliters of lower solution was transferred into the new centrifuge tube too, and a total of 400 μL of extracting solution was obtained. One hundred fifty microliters of extracting solution was transferred into a vial and dried. The residue was dissolved with 300 μL of a solution of isopropanol–methanol (*v:v*,1:1) and vortexed for 30 s. The solution was transferred into a 1.5 mL EP tube and centrifuged at 13,000 r/min for 10 min, and 150 μL supernatant was filtered through 0.22 μm microfilters into an LC vial. Quality control (QC) samples were prepared by mixing the same volume of extracting solution.

A Nexera UPLC (Shimadzu, Kyoto, Japan) with Waters ACQUITY UPLC BEH C_18_ (100 mm × 2.1 mm, 1.7 μm; Milford, MA, USA) was used to separate the extracts. The elution solution was consisted of (A) acetonitrile and water (*v:v*, 60:40), containing 0.1% formic acid and 10 mmol/L ammonium formate, and (B) acetonitrile and isopropyl alcohol (*v:v*, 10:90), containing 0.1% formic acid and 10 mmol/L ammonium formate. The elution program was as follows: 30% B over 0–3 min, 30–62% B over 3–5 min, 62–82% B over 5–15 min, 82–99% B over 15–16.5 min, 99% B over 16.5–18.0, 99–30% B over 18–18.1 min, 30% B over 18.1–20.0 min. The flow rate, column temperature and injection volume were 0.3 mL/min, 45 °C and 5 μL, respectively. A Q Exactive MS system (Thermo Scientific, Waltham, MA, USA) was operated using positive ion mode and negative ion mode heating electrospray ionization source (HESI). HESI source conditions were as follows: Positive mode: heater temp 300 °C, sheath gas flow rate 45 arb, aux gas flow rate 15 arb, sweep gas flow rate 1 arb, spray voltage 3.5 KV, capillary temp 320 °C, S-Lens RF level 50%. MS1 scan ranges: 120–1800. Negative mode: heater temp 300 °C, sheath gas flow rate 45 arb, aux gas flow rate 15 arb, sweep gas flow rate 1 arb, spray voltage 3.1 KV, capillary temp 320 °C, S-Lens RF level 50%. MS1 scan ranges: 120–1800. The mass–charge ratios of lipid molecules and lipid fragments were collected as follows: Ten fragment profiles were collected after each full scan. The resolution ratio of MS1 was 70,000 at *m/**z* 200, and the resolution ratio of MS2 was 17,500 at *m/**z* 200. The QCs were injected at regular intervals throughout the analytical run to provide the data for the repeatability of detecting system. The original Q Exactive LC-MS data in raw format were processed using the software Lipid Search. The molecular structure of lipids was identified according to the parent ions and multi-stage MS data. The results were aligned according to a certain retention time range and combined into a single report to sort out the original data matrix. In each sample, all peak signals were normalized (the signal intensity of each peak was converted to the relative intensity in the spectrum, then multiplied by 10,000). The extracted data were further processed by removing any peaks with a missing value (ion intensity = 0) in more than 50% of groups and by replacing the zero value with half of the minimum value. The positive and negative ion data were combined to form a data matrix.

### 2.5. Analysis of the Correlation of Lipids with Meat Quality of Yak Longissimus Dorsi (LD)

The volatile flavor compounds in livestock muscle are largely determined by the type and amount of lipids. Correlations between lipids and meat quality and between lipids and volatile flavor compounds were determined by Pearson correlation analysis in SPSS 16.0. Significant differences were considered at *p* < 0.05, and correlation coefficient > 0.8 or < −0.8 was considered as a high correlation. The important lipid molecules in yak LD being closely related to the a*, L*, shear force, pH, cooking loss and volatile flavor compounds of yak muscle were screened.

### 2.6. Statistical Analysis

The values of a*_45min_, b*_45min_, L*_45min_, a*_24h_, b*_24h_, L*_24h_, pH_45min_, pH_24h_, shear force, cooking loss, water loss and volatile flavor compounds and the total contents of aldehydes, ketones, alcohols, esters, aliphatic hydrocarbons, heterocyclic compounds, aromatics and acids were analyzed by independent-sample T-test in SPSS 16.0 (SPSS Inc, Chicago, IL, USA) and were represented as mean ± standard deviation (SD) of the mean. Principal component analysis (PCA) and orthogonal partial least squares discriminant analysis (OPLS-DA) were utilized to distinguish the lipids and volatile flavor compounds that were different in yak LD between the GF and SF groups using the ropls package in R version 3.6.2. To prevent overfitting, 7-fold cross-validation and 200 response permutation testing were used to evaluate the model quality. Variable importance of projection (VIP) values from the OPLS-DA were used to rank the overall contribution of each variable to group discrimination. A two-tailed Student’s T-test was further used to verify the significance of the differences in lipids in yak LD between the GF and SF groups. The Loading-PAC of volatile flavor compounds and significantly different lipids (SDLs) was determined by analyzing the volatile flavor compounds and SDLs in all 12 LD samples (GF and SF groups combined) using R version 3.6.2. Fold change (FC) was calculated by the ratio of the average relative lipid concentration in the LD of SF yaks to that of GF yaks. The variation tendency of lipid concentration in yak LD under different feeding systems can be judged by FC value. When FC was more than 1, the concentration of lipids in the LD of SF yaks was higher than that in the LD of GF yaks; when FC was less than 1, the concentration of lipids in the LD of SF yaks was lower than that in the LD of GF yaks.

## 3. Results

### 3.1. Meat Quality and Total Lipid Content of Yak LD

The values of meat quality of yak LD under GF and SF are shown in Table 1. The values of a*_45min_, a*_24h_ and L*_24h_ in the SF group were higher than those in the GF group (*p* < 0.01). The values of pH_45min_ and pH_24h_ in the SF group were lower than those in the GF group (*p* < 0.05). Moreover, the value of shear force in the SF group was lower than that in the GF group (*p* < 0.01), so the LD of SF yaks was more tender than that of GF yaks. The value of cooking loss in the SF group was lower than that in the GF group (*p* < 0.05). The total lipid content in the LD of SF yaks was higher that in the LD of GF yaks (*p* < 0.01).

### 3.2. Lipids in Yak LD

In total, 596 lipid molecules were simultaneously identified in yak LD under SF and GF (Appendix A). Among them, 512 lipids were detected in positive ion mode, whereas 84 lipids were found in negative ion mode. The classification of lipids is shown in Figure 1. These lipid molecules spanned over four major lipid categories, including 2 sterol lipids (STs), 51 sphingolipids (SPs), 225 glycerolipids (GLs) and 318 glycerophospholipids (GPs), which can be further attributed to 18 subclasses including 17 ceramides (Cers), 2 cholesteryl esters (ChEs), 37 diglycerides (DGs), 10 cardiolipins (CLs), 2 dimethylphosphatidylethanolamines (dMeEPEs), 39 lyso-phosphatidylcholines (LPCs), 1 lyso-dimethylphosphatidylethanolamines (LdMePEs), 5 lyso-phosphatidylethanolamines (LPEs), 2 monoglycerides (MGs), 2 phosphatidic acids (PAs), 163 phosphatidylcholines (PCs), 58 phosphatidylethanolamines (PEs), 1 phosphatidylglycerols (PGs), 13 phosphatidylinositols (PIs), 24 phosphatidylserines (PSs), 32 sphingomyelins (SMs), 2 sphingosines (SOs) and 186 triacylglycerols (TGs).

The score plots of PCA and OPLS-DA of lipids in yak LD under GF and SF are shown in Figure 2a,b, respectively. The PCA showed that the distribution of lipids in yak LD under SF and GF was completely divided into two spaces. The permutation test is a validation procedure that allows the comparison of the original results with those obtained under a scenario where no differences are expected between groups; it was used to randomly change the order of classification variables Y, and the values of R^2^ and Q^2^ in the random model were obtained by establishing multiple OPLS-DA models (*n* = 200). The results showed that R^2^Y(cum) was (0, 0.638) and Q^2^(cum) was (0, −0.734). The intercept between the regression line of Q^2^ and the vertical axis was less than 0, and the original model possessed better accuracy and robustness. The score scatterplot of OPLS-DA exhibited a total variance of 77.1%, of which component 1 was 55.6% and component 2 was 21.5%. The PCA and OPLS-DA showed that there were obvious differences in lipids between the GF and SF groups, and the different feeding systems induced a marked perturbation of lipids in yak LD. Further, 75 SDLs were identified in yak LD between GF and SF groups (Appendix A), including 2 dMeEPEs, 5 LPCs, 1 PAs, 22 PCs, 14 PEs, 2 PIs, 6 PSs, 5 SMs and 18 TGs. The relative concentrations of 26 SDLs in the LD of SF yaks were significantly higher than the values in the LD of GF yaks, including 2 dMeEPEs, 1 LPCs, 4 PCs, 6 PEs and 13 TGs, whereas the remaining 49 SDLs exhibited a decrease in SF group, including 4 LPCs, 1 PAs, 18 PCs, 8 PEs, 2 PIs, 6 PSs, 5 SMs and 5 TGs.

The Loading-PAC of SDLs in yak LD under GF and SF (Figure 2c) showed that the variance contribution rates of principal component 1 (PC1) and principal component 2 (PC2) were 75.52 and 11.25%, respectively, and the aggregate value was 86.77%. PE(18:0/22:4), PE(18:0/22:5), PC(18:2/18:0), TG(18:1/18:1/18:2), TG(18:0/18:1/18:1), PI(18:0/22:5), TG(16:0/16:1/18:1), TG(16:0/17:1/18:1), TG(16:1/18:1/18:1), PI(18:0/20:4), PE(18:1/18:1), PE(16:0/22:6) and TG(16:1/16:1/18:1) played an important role in the lipid differences between SF and GF yak LD from the angle of PC1; TG(17:0/18:1/18:1) and PE(18:0/22:5) played an important role in the lipid differences between SF and GF yak LD from the angle of PC2. The information on crucial lipids resulting in the lipid differences in yak LD between GF and SF is shown in Table 2.

### 3.3. Volatile Flavor Compounds in Yak LD

The score plots of PCA and OPLS-DA of volatile flavor compounds in yak LD under GF and SF are shown in Figure 3a,b, respectively. The PCA showed that the distribution of volatile flavor compounds in the individual samples of yak LD under GF and SF was completely divided into two spaces. The permutation test for OPLS-DA model (*n* = 200) showed that R^2^Y(cum) was (0, 0.64) and Q^2^(cum) was (0, −0.27). Therefore, the original model possessed better accuracy and robustness. The score plots of PCA and OPLS-DA of volatile flavor substances in yak LD under different feeding systems showed that there were obviously differences in volatile flavor compounds between SF and GF yak LD, and the feeding system for yak can greatly affect the volatile flavor compounds in yak LD.

As shown in Table 3, in total, 52 volatile flavor compounds were detected in yak LD. These volatile flavor compounds included 15 aldehydes, 6 ketones, 10 alcohols, 3 esters, 8 aliphatic hydrocarbons, 2 heterocyclic compounds, 4 aromatics and 4 acids. Among them, 49 (12 aldehydes, 6 ketones, 10 alcohols, 3 ester, 8 aliphatic hydrocarbons, 4 aromatics, 2 heterocyclic compounds and 4 acids) were detected in the GF group and 39 (11 aldehydes, 6 ketones, 8 alcohols, 3 esters, 3 aliphatic hydrocarbons, 3 aromatics, 2 heterocyclic compounds and 3 acids) were detected in SF group.

The main volatile flavor compounds in both GF and SF groups were pelargonic aldehyde, 3-hydroxy-2-butanone and 1-octen-3-ol. The composition of various volatile flavor compounds in the yak LD under GF and SF is shown in Figure 4a,b, respectively. The highest content in both GF and SF groups was aldehydes, followed by alcohols and ketones in sequence. The total contents of aldehydes, alcohols and esters in SF group were higher than the values in the GF group (*p* < 0.05), whereas ketones, heterocyclic compounds and acids were lower (*p* < 0.05). The Loading-PAC of volatile flavor compounds in yak LD is shown in Figure 4c. The variance contribution rates of PC1 and PC2 were 67.54% and 8.81%, respectively, and the aggregate value was 76.35%. The compounds 1-heptanol (F27), vinyl acetate (F34), acetophenone (F21), 2-heptanone (F19), 3-hydroxy-2-butanone (F18), 1-octanol (F29), 1-pentanol (F24), ethyl caprate (F32), pentylcyclopropane (F41), pelargonic aldehyde (F6) and 1-octen-3-ol (F30) played an important role in the flavor differences between SF and GF yak LD from the angle of PC1; heptaldehyde (F4) and 2,2,4,4,6,8,8-heptamethylnonane (F37) played an important role in the flavor differences between SF and GF yak LD from the angle of PC2.

### 3.4. Results of Correlations

The correlations of crucial lipids with meat quality and volatile flavor compounds are shown in Figure 5a,b. Shear force was negatively correlated with the concentrations of TG(16:1/16:1/18:1), TG(16:1/18:1/18:1), TG(16:0/17:1/18:1) and TG(16:0/16:1/18:1); the cooking loss was positively correlated with the concentration of PE(18:0/22:5); a*_45min_ and a*_24h_ were highly positively correlated with the concentrations of PE(18:0/22:4), PC(18:2/18:0), whereas they were negatively correlated with the concentration of PE(18:0/22:5); L*_24h_ was highly positively correlated with the concentrations of PE(18:0/22:4), PC(18:2/18:0), TG(18:1/18:1/18:2), TG(18:0/18:1/18:1), TG(16:0/16:1/18:1), TG(16:0/17:1/18:1) and TG(16:1/18:1/18:1). On the other hand, total content of aldehydes was highly positively correlated with the concentrations of PC(18:2/18:0), TG(16:1/16:1/18:1) and TG(16:0/16:1/18:1); the total content of ketones was highly positively correlated with the concentrations of PE(18:0/22:5), PI(18:0/20:4), PI(18:0/22:5) and PC(18:2/18:0); and the total content of alcohols was highly positively correlated with the concentrations of TG(16:1/18:1/18:1), PE(18:0/22:4), PC(18:2/18:0), TG(16:0/16:1/18:1) and TG(16:1/16:1/18:1).

## 4. Discussion

The lipids in meat are crucial in beef quality evaluation [34], and the lipid content in muscle can affect many indexes of meat quality [35]. Compared with ordinary beef, snowflake beef contains higher intramuscular lipid content, and the lipid oxidation in snowflake beef is more intense, which may aggravate myoglobin oxidation and affect the meat color [36]. The total lipid content in the LD of SF yaks was higher than the value in the LD of SF yaks, and the values of a*_45min_, a*_24h_ and L*_24h_ in the LD of SF yaks were higher than the values in the LD of GF yaks too. Therefore, the lipids in muscle can directly affect the a* and L*_24h_ of yak muscle. The shear force, which reflects the palatability and chewiness of meat, is an important parameter measuring meat tenderness and is negatively correlated with muscle tenderness [37,38]. The increase in intramuscular lipid content can cut off the connecting structure among muscle fibers, thus reducing shear force and improving meat tenderness. With the increase in lipid content, beef tenderness is gradually improved [39]. The increase in the lipid content in yak muscle also can improve yak muscle tenderness. In this study, the a* value of yak LD was positively correlated with the concentrations of PE(18:0/22:4) and PC(18:2/18:0) in yak LD, and the L*_24h_ value of yak LD was positively correlated with the concentrations of PE(18:0/22:4), PC(18:2/18:0), TG(18:1/18:1/18:2), TG(18:0/18:1/18:1), TG(16:0/16:1/18:1), TG(16:0/17:1/18:1) and TG(16:1/18:1/18:1); shear force was negatively correlated with the concentrations of TG(16:1/16:1/18:1), TG(16:1/18:1/18:1), TG(16:0/17:1/18:1) and TG(16:0/16:1/18:1). In terms of meat color and tenderness, consumers much prefer meat with a high a* value and low L* and shear force values. Therefore, TG(16:1/18:1/18:1), TG(16:0/17:1/18:1), TG(16:0/16:1/18:1), PE(18:0/22:4) and PC(18:2/18:0) in yak muscle play a major role in improving the a* and tenderness of yak meat.

Volatile flavor compounds are mainly produced by lipid oxidation and degradation, and many studies show beef flavor is directly related to lipids. Lipids can decompose and produce fatty acids, and then these fatty acids transfer into specific flavor compounds. The meat of lamb and Argentine beef fed with grass possessed more kinds of volatile flavor substances than the meat of those fed with grain-based diets [40,41]. In this study, there were 49 volatile flavor substances in the LD of yaks fed with grass, whereas there were 39 volatile flavor compounds in the LD of yaks fed with TMR. The muscle of yaks fed with grass possessed more kinds of volatile flavor compounds too. It can be inferred that grazing yaks have greater exercise and ingest more kinds of fatty acids and minerals from grass. These compounds can produce more kinds of volatile flavor substances in yak muscle. Because UFAs are easily oxidized, the composition of volatile flavor compounds in livestock is greatly affected by UFAs. UFAs can generate hundreds of volatile flavor compounds, including hydrocarbons, aldehydes, alcohols, ketones, esters and heterocyclic compounds [42,43]. The higher the content of UFAs, the stronger the flavor [11]. The fatty acids including C16:1, C18:1, C18:2, C18:3 and C18:4 were the leading UFAs in yak muscle, so the lipids derived from C16:1, C18:1, C18:2 and C18:3 played a key role in the regulation of flavor in yak muscle. These crucial lipids related to flavor in yak muscle included TG(18:1/18:1/18:2), TG(18:0/18:1/18:1), PE(18:1/18:1), TG(16:0/16:1/18:1), PC(18:2/18:0), TG(16:0/17:1/18:1), TG(16:1/18:1/18:1), PI(18:0/20:4), TG(16:1/16:1/18:1) and TG(17:0/18:1/18:1), and the concentrations of these lipids in the SF group were all higher than those in the GF group. Therefore, the yak muscle in the SF group was likely to possess a stronger smell.

Aldehydes possess a very low aroma threshold value [44] and can cause a sweet and fruity smell. They are very important characteristic aroma components in beef [45] and are the major degradation products of UFAs [46]. UFAs can produce hexanal, glutaraldehyde, nonanal, heptanaldehyde, octanal, 2-octenal, 2-decenenal, 2-hexenal and so on in goat meat [47]. Hexanal, heptanaldehyde, octanal and nonanal are related to the oxidative degradation of C18:1, C18:2 and C20:4 [41,48]. Decanal is related to the oxidative degradation of C18:1, and 2,4-decene aldehyde is related to the oxidative degradation of C18:2 in lamb meat [49]. The total content of aldehydes in yak muscle is the highest among all volatile flavor compounds, so aldehydes were the leading volatile flavor compounds in yak muscle. The content of pelargonic aldehyde in both GF and SF groups was the highest among all aldehydes. Pelargonic aldehyde is mainly derived from the C18:1 in lipids, and it was highly positively correlated with TG(16:0/16:1/18:1), TG(16:1/16:1/18:1), TG(16:1/18:1/18:1) and TG(16:0/17:1/18:1). From the correlation results, it can be found the change in aldehydes in yak muscle was mainly caused by PC(18:2/18:0), TG(16:1/16:1/18:1), TG(16:0/16:1/18:1), TG(16:1/16:1/18:1), TG(16:1/18:1/18:1) and TG(16:0/17:1/18:1). Ketones are the products of lipid oxidation [50,51], and most ketones possess a creamy or fruity aroma. The content of 3-hydroxy-2-butanone coming from the degradation of many fatty acids in yak muscle was the highest among all ketones. Studies showed that heptanone, octanone and 2,3-octanedione were related to the degradation of C18:2 [52,53]. From the correlation results, it can be found that PC(18:2/18:0), PI(18:0/20:4) and PI(18:0/22:5) were the most important factors that affected the ketones in yak muscle. Alcohols in livestock meat are mainly produced by the degradation of conjugated C18:2 and C18:1. Hexanol comes from the degradation of C16:1 and C18:1, and pentanol and octanol are related to the degradation of C18:2 and C18:1, respectively [50]. One-octene-3-ol in beef is related to the degradation of C18:2 [54], and the content of 1-octen-3-ol in yak muscle was the highest among all alcohols. From the correlation results, it can be inferred the effects of lipids on alcohols in yak muscle were mainly realized by the change in PC(18:2/18:0), TG(16:0/16:1/18:1), TG(16:1/18:1/18:1) and TG(16:1/16:1/18:1).

## 5. Conclusions

The leading lipids in yak muscle under both SF and GF were SPs, GLs and GPs, and the leading volatile flavor compounds in yak muscle under both SF and GF were aldehydes, alcohols and ketones. The contents of pelargonic aldehyde, 3-hydroxy-2-butanone and 1-octen-3-ol were the highest among all volatile flavor compounds. By contrast with the muscle of SF yak, the volatile flavor compounds in the muscle of GF yak were more diversified. The effects of lipids on the a* of yak muscle were mainly realized by PE(18:0/22:4), and the effects of lipids on the tenderness of yak muscle were mainly realized by PC(18:2/18:0), TG(16:1/18:1/18:1), TG(16:0/17:1/18:1) and TG(16:0/16:1/18:1). The high content of the above lipids can improve the yak meat quality. The change in volatile flavor compounds in yak muscle was mainly caused by the change in the content of TG(18:1/18:1/18:2), TG(18:0/18:1/18:1), PI(18:0/20:4), TG(16:0/17:1/18:1), TG(16:0/16:1/18:1), PC(18:2/18:0), TG(16:1/18:1/18:1), TG(16:1/16:1/18:1) and TG(17:0/18:1/18:1). Of these lipids, PC(18:2/18:0) was the most critical factor regulating the volatile flavor compounds, including aldehydes, alcohols and ketones, in yak meat.

## Figures and Tables

**Figure 1 animals-12-02814-f001:**
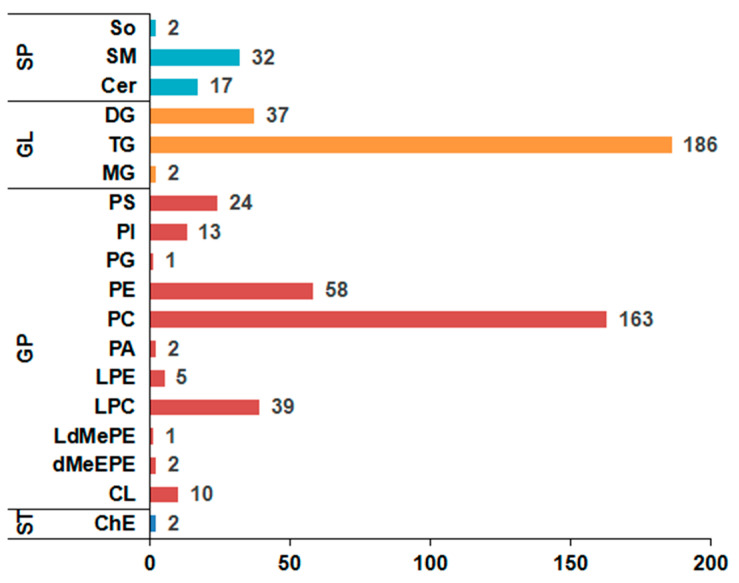
The column chart of the lipid classification in yak longissimus dorsi (LD). Column length represents the number of lipids. ST represents sterol lipid; SP represents sphingolipid; GL represents glycerolipid; GP represents glycerophospholipid; Cer represents ceramide; ChE represents cholesteryl ester; DG represents diglyceride; CL represents cardiolipin; dMeEPE represents dimethylphosphatidylethanolamine; LPC represents lyso-phosphatidylcholine; LdMePE represents lyso-dimethylphosphatidylethanolamine; LPE represents lyso-phosphatidylethanolamine; MG represents monoglyceride; PA represents phosphatidic acid; PC represents phosphatidylcholine; PE represents phosphatidylethanolamine; PG represents phosphatidylglycerol; PI represents phosphatidylinositol; PS represents phosphatidylserine; SM represents sphingomyelin; So represents sphingosine; TG represents triacylglycerol.

**Figure 2 animals-12-02814-f002:**
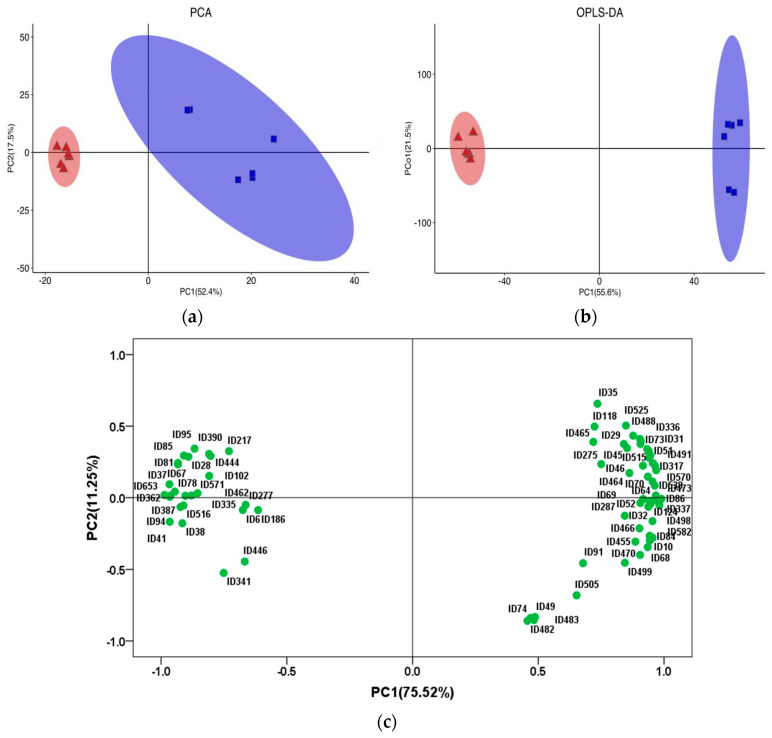
(**a**) The score plot of principal component analysis (PCA) of lipids in yak LD under graze feeding (GF) and stall feeding (SF). Red represents SF group; blue represents GF group. (**b**) The score plot of orthogonal partial least squares discriminant analysis (OPLS-DA) of lipids in yak LD under GF and SF. (**c**) Loading-PAC of significantly different lipids (SDLs) in the LD of GF and SF yaks. PC1 represents principal component 1, and PC2 represents principal component 2.

**Figure 3 animals-12-02814-f003:**
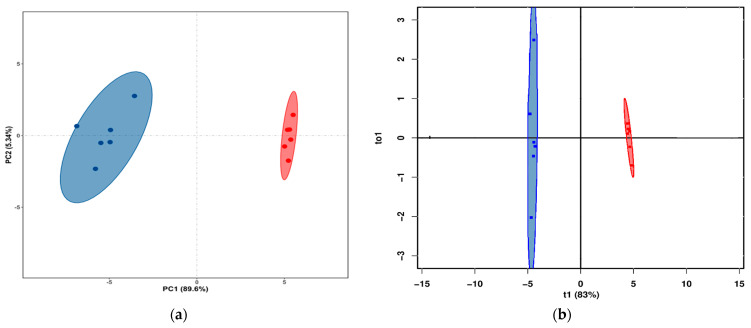
(**a**) The PCA score plot of volatile flavor compounds in yak LD under GF and SF. Different colors represent different groups. Blue represents the samples in the GF group, and red represents the samples in the SF group. (**b**) The result of OPLS-DA analysis of the volatile flavor compounds in yak LD under GF and SF. Blue represents the samples in the GF group, and red represents the samples in the SF group.

**Figure 4 animals-12-02814-f004:**
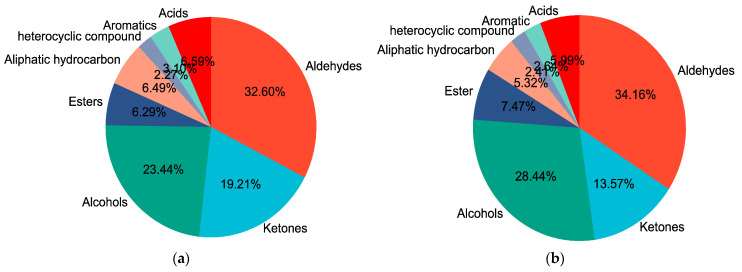
(**a**) The composition of various volatile flavor compounds in the yak LD under GF. (**b**) The composition of various volatile flavor compounds in the yak LD under SF. (**c**) The Loading-PAC of volatile flavor compounds in all yak LD samples.

**Figure 5 animals-12-02814-f005:**
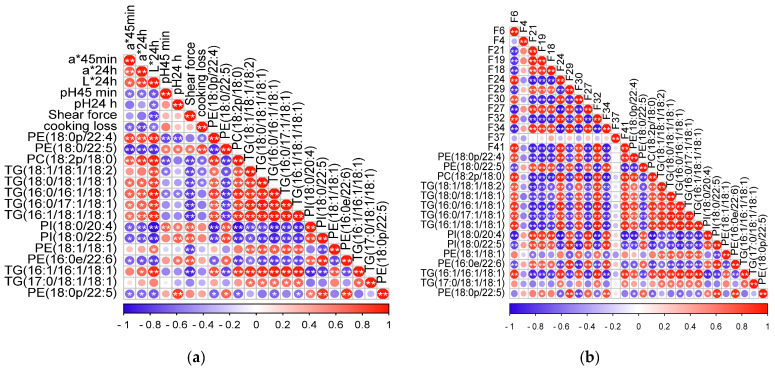
(**a**) The correlations of crucial lipids with meat quality. (**b**) The correlations of crucial lipids with volatile flavor compounds. * shows *p* < 0.05, and ** shows *p* < 0.01. Circles with no * or ** show *p* > 0.05. Red and blue represent positive and negative correlations, respectively. The deeper the color, the higher the correlation.

**Table 1 animals-12-02814-t001:** The meat quality and total lipid content of yak longissimus dorsi (LD) under graze feeding (GF) and stall feeding (SF).

Variable	GF Group (Mean ± SD, *n* = 6)	SF Group (Mean ± SD, *n* = 6)
a^*^_45min_	29.10 ± 0.29	30.67 ± 0.99 **
b^*^_45min_	6.69 ± 0.07	6.45 ± 0.23
L^*^_45min_	6.93 ± 0.39	7.27 ± 0.24
a^*^_24h_	31.79 ± 0.46	33.49 ± 0.64 **
b^*^_24h_	9.07 ± 0.36	8.74 ± 0.19
L^*^_24h_	7.31 ± 0.04	8.55 ± 0.07 **
pH_45min_	6.45 ± 0.11	6.28 ± 0.07 *
pH_24h_	5.76 ± 0.05	5.65 ± 0.09 *
Shear force (kg)	17.13 ± 0.75	14.77 ± 0.01 **
Cooking loss (%)	30.51 ± 0.91	28.24 ± 1.88 *
Water loss rate (%)	20.53 ± 1.03	19.50 ± 1.01
Total lipids content (%)	1.73 ± 0.10	2.71 ± 0.11 **

GF represents graze feeding; SF represents stall feeding; SD represents standard deviation; L*, a* and b* represent lightness, redness and yellowness, respectively; * shows *p* < 0.05, and ** shows *p* < 0.01.

**Table 2 animals-12-02814-t002:** The information on crucial lipids resulting in the lipid differences in yak LD between GF and SF groups.

Lipid	ID	Class	Formula	Fatty Acid	FC
PE(18:0/22:4)	86	PE	C_45_ H_80_ O_7_ N_1_ P_1_	C18:0, C22:4	3.09
PE(18:0/22:5)	147	PE	C_45_ H_78_ O_8_ N_1_ P_1_	C18:0, C22:5	0.56
PC(18:2/18:0)	337	PC	C_44_ H_84_ O_7_ N_1_ P_1_	C18:2, C18:0	2.33
TG(18:1/18:1/18:2)	84	TG	C_57_ H_102_ O_6_	C18:1, C18:1, C18:2	2.65
TG(18:0/18:1/18:1)	51	TG	C_57_ H_106_ O_6_	C18:0, C18:1, C18:1	1.95
TG(16:0/16:1/18:1)	37	TG	C_53_ H_98_ O_6_	C16:0, C16:1, C18:1	3.70
TG(16:0/17:1/18:1)	41	TG	C_54_ H_100_ O_6_	C16:0, C17:1, C18:1	2.43
TG(16:1/18:1/18:1)	94	TG	C_55_ H_100_ O_6_	C16:1, C18:1, C18:1	3.54
PI(18:0/20:4)	78	PI	C_47_ H_81_ O_13_ P_1_	C18:0, C20:4	0.43
PI(18:0/22:5)	74	PI	C_49_ H_83_ O_13_ P_1_	C18:0, C22:5	0.13
PE(18:1/18:1)	67	PE	C_41_ H_76_ O_8_ N_1_ P_1_	C18:1, C18:1	1.84
PE(16:0/22:6)	387	PE	C_43_ H_74_ O_7_ N_1_ P_1_	C16:0, C22:6	0.61
TG(16:1/16:1/18:1)	38	TG	C_53_ H_96_ O_6_	C16:1, C16:1, C18:1	3.25
TG(17:0/18:1/18:1)	49	TG	C_56_ H_104_ O_6_	C17:0, C18:1, C18:1	1.96
PE(18:0/22:5)	81	PE	C_45_ H_78_ O_7_ N_1_ P_1_	C18:0, C22:5	0.66

FC represents fold change, which is the ratio of the average relative lipid concentration in the LD of SF yaks to that of GF yaks; PE represents phosphatidylethanolamine; PC represents phosphatidylcholine; TG represents triacylglycerol; PI represents phosphatidylinositol.

**Table 3 animals-12-02814-t003:** The relative content of volatile flavor compounds in the yak LD under SF and GF.

Volatile Flavor Compound	GF Group (Mean ± SD, *n* = 6)	SF Group (Mean ± SD, *n* = 6)
Acetaldehyde (F1)	2.91 ± 0.29	2.92 ± 0.11
3-methyl butyraldehyde (F2)	1.68 ± 0.12	ND
Caproaldehyde (F3)	1.91 ± 0.15	2.26 ± 0.14 **
Heptaldehyde (F4)	2.31 ± 0.30	2.06 ± 0.08
Caprylaldehyde (F5)	1.83 ± 0.33	2.10 ± 0.09
Pelargonic aldehyde (F6)	12.59 ± 0.89	17.39 ± 0.84 **
Capric aldehyde (F7)	0.18 ± 0.03	0.29 ± 0.02 **
2-nonene aldehyde (F8)	ND	1.27 ± 0.06
2-decyl olefine aldehyde (F9)	2.16 ± 0.16	ND
2-undecenal (F10)	1.61 ± 0.18	ND
Myristic aldehyde (F11)	ND	1.52 ± 0.06
Hexadecanal (F12)	3.10 ± 0.48	3.34 ± 0.10
Stearic aldehyde (F13)	1.63 ± 0.13	ND
Benzaldehyde (F14)	0.31 ± 0.07	0.22 ± 0.02 *
3-ethyl benzaldehyde (F15)	ND	0.51 ± 0.04
Acetone (F16)	1.40 ± 0.10	1.56 ± 0.08 *
2-butanone (F17)	0.57 ± 0.06	0.61 ± 0.05
3-hydroxy-2-butanone (F18)	10.80 ± 0.92	7.31 ± 0.61 **
2-heptanone (F19)	1.53 ± 0.08	1.03 ± 0.11 **
6-methyl-5-hepten-2-one (F20)	0.73 ± 0.16	0.67 ± 0.05
Acetophenone (F21)	3.95 ± 0.42	2.27 ± 0.15 **
Ethyl alcohol (F22)	0.83 ± 0.10	1.44 ± 0.14 **
3-methyl-1-butanol (F23)	1.78 ± 0.06	ND
1-pentanol (F24)	2.66 ± 0.13	4.57 ±0.14 **
3-methyl-2-buten-1-ol (F25)	0.76 ± 0.06	0.58 ± 0.08 **
1-hexanol (F26)	2.09 ± 0.15	2.43 ± 0.15 **
1-heptanol (F27)	2.78 ± 0.11	1.82 ± 0.16 **
2-ethylhexanol (F28)	2.10 ± 0.17	2.49 ± 0.17 **
1-octanol (F29)	2.19 ± 0.22	1.43 ± 0.14 **
1-octen-3-ol (F30)	7.83 ± 1.51	13.40 ± 0.57 **
2-hexadecanol (F31)	0.14 ± 0.03	ND
Ethyl caprate (F32)	2.33 ± 0.15	5.20 ± 0.28 **
N-decyl Acetate (F33)	0.10 ± 0.02	ND
Vinyl acetate (F34)	3.04 ± 0.29	1.46 ± 0.14 **
Vinyl Hexanoate (F35)	0.75 ± 0.09	0.73 ± 0.06
Octadecane (F36)	0.51 ± 0.04	ND
2,2,4,4,6,8,8-heptamethylnonane (F37)	2.60 ± 0.40	2.44 ± 0.25
n-tridecane (F38)	0.10 ± 0.03	0.05 ± 0.01 **
DL-limonene (F39)	0.81 ± 0.08	ND
Decane (F40)	0.27 ± 0.05	ND
Pentylcyclopropane (F41)	1.53 ± 0.18	2.78 ± 0.14 **
Styrene (F42)	0.60 ± 0.09	ND
2-ethylfuran(F43)	0.56 ± 0.08	0.82 ± 0.07 **
2-pentylfuran (F43)	1.69 ± 0.10	1.56 ± 0.13
Toluene (F45)	0.79 ± 0.08	0.90 ± 0.08
Phenol (F46)	0.30 ± 0.14	ND
Naphthalene (F47)	0.81 ± 0.29	1.16 ± 0.20
m-xylene (F48)	1.16 ± 0.21	0.55 ± 0.09 **
Acetic acid (F49)	2.21 ± 0.21	2.75 ± 0.16
Butyric acid (F50)	1.41 ± 0.19	1.40 ± 0.15
4-hydroxybutyric acid (F51)	0.49 ± 0.10	ND
Hexanoic acid (F52)	2.40 ± 0.29	1.78 ± 0.16 **

* represents *p* < 0.05 and ** represents *p* < 0.01 compared with the GF group. ND represents no determination.

## Data Availability

The original contributions presented in the study are included in the article and the Appendix A; further inquiries can be directed to the corresponding authors.

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
