# Peer review of "Effect of Lipids in Yak Muscle under Different Feeding Systems on Meat Quality Based on Untargeted Lipidomics"

_animals, 2022, doi:10.3390/ani12202814_

Round 1

Reviewer 1 Report (Previous Reviewer 1)

Comments in the attached file.

Author Response

Reviewer 2 Report (Previous Reviewer 2)

I only have one commment, I would remove the sensory word as meat quality, because they did not do sensory analysis.

Additional comments:

I think the article is well written, it is interesting, I liked the authors' response a lot because they showed that they worked on what the reviewers marked.

Some additional questions: I think the article could be more complete if a sensory evaluation was added, since this is the best method to measure taste.

I agree with the answer that the Longissimus dorsi is the most used muscle when texture is measured, if there are differences with other muscles of the carcass, especially in fat.

I think the article has a very good quality to be published in the journal. The only detail that I see is that yak meat is not consumed all over the world, so it becomes a local issue.

Round 2

Reviewer 1 Report (Previous Reviewer 1)

See attached file

Author Response

This manuscript is a resubmission of an earlier submission. The following is a list of the peer review reports and author responses from that submission.

Round 1

Reviewer 1 Report

Described in the attached file

Reviewer 2 Report

The article is interesting but I think it needs to explain several things.

Using 20 animals and divided them into 2 treatments of 6 animals each. Where are the remaining 8?

It would be important to know how the type of feeding affected the final weight of the animals.

Why did they only sample the LD and not do these studies in any other muscle?

There are variations between different muscles.

I understand that the article is about the effect on lipids, however the authors carry out different physicochemical analyzes of meat quality, but in the end there is no discussion of these results.

The results showed that the animals fed with GF had 49 volatile substances and those fed with SF had 39. In the discussion it is never discussed why this was.

In your discussion you put that a high content of lipids can affect softness, but your results showed the opposite, less lipids greater softness, how do you explain it?
